# Characteristics of 5015 Salivary Gland Neoplasms Registered in the Hiroshima Tumor Tissue Registry over a Period of 39 Years

**DOI:** 10.3390/jcm8050566

**Published:** 2019-04-26

**Authors:** Kazuhiro Sentani, Ikuko Ogawa, Kotaro Ozasa, Atsuko Sadakane, Mai Utada, Takafumi Tsuya, Hiroki Kajihara, Shuji Yonehara, Yukio Takeshima, Wataru Yasui

**Affiliations:** 1Department of Molecular Pathology, Hiroshima University Graduate School of Biomedical and Health Sciences, Hiroshima 734-8551, Japan; wyasui@hiroshima-u.ac.jp; 2Center of Oral Clinical Examination, Hiroshima University Hospital, Hiroshima 734-8551, Japan; dlabo@hiroshima-u.ac.jp; 3Department of Epidemiology, Radiation Effects Research Foundation, Hiroshima 732-0815, Japan; ozasa@rerf.or.jp (K.O.); sadakane@rerf.or.jp (A.S.); utada@rerf.or.jp (M.U.); 4Hiroshima Prefecture Medical Association, Hiroshima 732-0057, Japan; takafumi.tsuya@gmail.com (T.T.); kajihara@hiroshima.med.or.jp (H.K.); 5Department of Pathology and Research Laboratory, Welfare Association Onomichi General Hospital, Onomichi, Hiroshima 722-8508, Japan; yonehara@eos.ocn.ne.jp; 6Department of Pathology, Hiroshima University Graduate School of Biomedical and Health Sciences, Hiroshima 734-8551, Japan; ykotake@hiroshima-u.ac.jp

**Keywords:** salivary gland tumors, Japanese population, epidemiology, histology, time trends

## Abstract

Salivary gland neoplasms are uncommon, and their epidemiology in Japan has not been well described. We conducted a retrospective review of salivary gland tumors registered in the Hiroshima Tumor Tissue Registry over a period of 39 years. The subjects were 5015 cases ranging in age from 6 to 97 (mean, 54.3) years old. The incidence of both benign tumors and malignant tumors increased with age until 60–69 years and then declined. Among the 5015 salivary gland neoplasms, 3998 (80%) were benign and 1017 (20%) were malignant. Pleomorphic adenoma (PA) was the most frequent benign tumor (68%), followed by Warthin tumor (26%). Adenoid cystic carcinoma (ACC) (27%) and mucoepidermoid carcinoma (MEC) (26%) were the two most frequent malignant tumors. Characteristically, there was a very low incidence of polymorphous adenocarcinoma in Japan. The average annual age-adjusted incidence rate per 100,000 population was 3.3 for benign tumors and 0.8 for malignant tumors. This is the large-scale multi-institutional analysis to describe the characteristics of salivary gland neoplasms, based on the pathological tissue registry data. We hope that the present data can contribute to early diagnosis and effective treatment of salivary gland tumors and to cancer prevention.

## 1. Introduction

Salivary gland tumors are relatively rare neoplasms accounting for 3–10% of all head and neck tumors, with an estimated annual incidence of 0.4–13.5 new cases per 100,000 population [1,2]. The standardized incidence of rates of malignant neoplasm derived from major salivary glands and minor salivary glands was reported to be 1.04 per 100,000 in the Japanese population and was slightly less than that of 1.31 in the European Union [3].

Despite their rarity, salivary gland neoplasms represent a wide variety of benign and malignant histological subtypes that is apparently unparalleled in comparison to any other organs. Although previous epidemiologic and demographic studies on salivary gland tumors from various parts of the world [4,5,6,7,8,9] have provided valuable knowledge, there were also discrepancies in these tumors between different geographic areas and ethnic groups [10,11]. In addition, some of the data became contradictory and less precise for several reasons, such as the transition in classification over a few decades, the omission of the registration of benign tumors, which constitute the majority of salivary gland tumors [12], and the inter-institutional bias of tumors in cancer institutions or referral centers for problematic cases, as mentioned by Buchner et al. [4]. The best source from which to obtain information on the true relative frequency of tumors is the records of a large-scale multi-institutional registration data covering a wide area.

The Hiroshima Tumor Tissue Registry (HTTR) is a population-based pathological tissue registry that was started in 1973 by the Hiroshima Prefecture Medical Association with the cooperation of the Radiation Effects Research Foundation (RERF). It covers the entire Hiroshima Prefecture (population in 2012: 2.86 million, 2.6% of the Japanese population) in Japan. Pathologists routinely register the pathological diagnosis of all tumors and provide representative slides of resected malignant tumors, including biopsy specimens. When there are multiple tumor records in the same patient, the pathologist panels decide whether the records are reports of the same primary tumor or of multiple primary tumors by looking into pathological reports or performing a microscopic examination, if necessary. The HTTR has provided all data on malignant tumors to the Hiroshima Prefecture Cancer Registry (HPCR) and has contributed to improving the quality of its data since its inception in 2002.

To date, there are few reports in the English literature on the epidemiology of salivary gland tumors in the Japanese population. Thus, the aim of this study was to clarify the relative frequency, characteristics and time trends of various salivary gland tumors in a Japanese population by using large-scale multi-institutional data that were registered in the HTTR over a period of 39 years in Hiroshima Prefecture, Japan.

## 2. Materials and Methods

Patients were all of the cases diagnosed as having primary salivary gland tumors that were registered in the HTTR from 1973 to 2011, which was the most recent follow-up period when we started the present study. In 2011, 85 institutions, including all designated cancer care hospitals, hospitals, clinics, and clinical laboratories, reported their tumor diagnoses to the HTTR. All metastatic neoplasms from other organs including the head and neck areas were excluded. Though tumors were originally coded according to the International Classification of Disease for Oncology (ICD-O), first, second, or third revision, all of the major and minor salivary gland tumors were re-grouped according to ICD-O-3 topography and morphology codes in the present study. All of the 1017 malignant tumors were histologically reviewed and reclassified independently by two pathologists (K.S. and I.O.) based on the latest (4th edition) of the World Health Organization (WHO) classification [1]. The concordance rate for diagnosis between two pathologists showed more than 80%, and when the evaluations differed, a final diagnosis was made by consensus while the investigators reviewed the specimen with a multi-head microscope. Newly described or re-defined entities, such as secretory carcinoma (SC), polymorphous adenocarcinoma, or poorly differentiated carcinoma, were diagnosed based on morphological and immunohistochemical findings, although the immunohistochemical or translocation study were not performed in all of these cases. All malignant tumors that show histological evidence of arising from pleomorphic adenoma (PA) were classified as carcinoma ex PA, although histological type of the malignant component were not described in detail. We described the number and the proportion of cases by behavior (benign or malignant), sex, site of tumor occurrence, histological type, period at diagnosis (in five-year groups), and age of diagnosis (in 10-year groups). We presented temporal trends (5-year calendar period) in sex-specific age-standardized registration rate per 100,000 population (standardized by Segi’s World Standard Population) by behavior (benign or malignant) and histological type. No formal statistical tests were performed. The present study was approved by the Data Usage Committee of the HTTR. It was conducted by the HTTR Working Committee and supported by the Hiroshima Prefecture Medical Association and the RERF.

## 3. Results

### 3.1. General Features and Time Trends by Sex, Age, Tumor Locations, and Histological Types

Of the 5015 salivary gland tumors from 2514 males and 2501 females, 3998 (80%) were benign and 1017 (20%) were malignant, and the ratios of benign to malignant tumors (B/M) were equally 4:1 in both sexes (Figure 1). The tumor locations were derived from 4318 (86%) major salivary glands and 697 (14%) minor salivary glands. In the males, there were 2224 major salivary gland tumors, with 1855 (83%) being benign and 369 (17%) malignant, and 290 minor salivary gland tumors, with 150 (52%) being benign and 140 (48%) malignant. In the females, there were 2094 major salivary gland tumors, with 1771 (85%) being benign and 323 (15%) malignant, and 407 minor salivary gland tumors, with 222 (55%) being benign and 185 (45%) malignant (Figure 1). More than 80% of the major salivary gland tumors were benign, whereas 40-50% of the minor salivary gland tumors were malignant (Table 1 and Table 2; Figure 2). Histologically, pleomorphic adenoma (PA) was the most frequent benign tumor (68%), followed by Warthin tumor (26%) and basal cell adenoma (3%). Adenoid cystic carcinoma (ACC) (27%) and mucoepidermoid carcinoma (MEC) (26%) were the two most frequent malignant tumors, followed by carcinoma ex PA (11%), adenocarcinoma, not otherwise specified (NOS) (7%), malignant lymphomas (6%) and acinic cell carcinoma (AcCC) (6%) (Figure 3; Appendix A).

The cases ranged in age from 6 to 97, with a mean age of 54.3 years old and a median age of 57 years old. The peak incidence was in the seventh decade (26%, 660/2514) in the males and in the sixth (20%, 504/2501) and seventh decades (20%, 503/2501) in the females. The mean and median ages of the cases presenting with benign tumors were 52.9 and 55 years old, and those with malignant tumors were 59.8 and 63 years old, respectively. The incidence of both benign tumors and malignant tumors increased with age until 60–69 years and then declined (Table 1).

In major salivary glands, benign tumors outnumbered malignant tumors in every decade of life, and there was a slight male predominance except for benign tumors in cases aged 10–49 years and malignant tumors in cases aged 20–29 years. In minor salivary glands, benign tumors were larger in number than malignant tumors until the fifth decade, and malignant tumors outnumbered benign tumors in the older age groups. In every decade of life, there was a female predominance in both benign and malignant tumors (Table 1). Salivary gland neoplasms are rare in children: 87% (20/23) of the tumors from children younger than 12 years of age were benign, with PA (90%, 18/20) being the most common, followed by lymphangioma (10%, 2/20). Malignant tumors in children consisted of MEC (33%, 1/3), adenocarcinoma, NOS (33%, 1/3), and Burkitt lymphoma (33%, 1/3). For teenagers aged 13–19 years, 85% (100/117) were benign, with PA (97%, 97/100) being the most common, followed by Warthin tumor (3%, 3/100). Malignant tumors in teenagers consisted of MEC (41%, 7/17), secretory carcinoma (SC) (24%, 4/17), AcCC (18%, 3/17), ACC (12%, 2/17), and carcinoma ex PA (6%, 1/17) (Table 1; Appendix A; Figure 4).

Among the salivary gland neoplasms, the most common site of major salivary gland tumors was the parotid gland, accounting for 83% (3016/3626) of the benign and 71% (489/692) of the malignant major salivary gland tumors. Tumors of the submandibular gland and sublingual gland accounted for 16% (576/3626) and 0.2% (8/3626) of the benign and 25% (171/692) and 3% (24/692) of the malignant major salivary gland tumors, respectively. Only the submandibular gland site showed a female predominance in both benign and malignant major salivary gland tumors, although all other sites of the major salivary glands displayed a male predominance (Table 2). Distinguished from other major salivary glands, 68% (13/19) of tumors in the sublingual gland from males and 85% (11/ 13) from females were malignant (Figure 2). The next most common location of tumor involvement among minor salivary glands was the palatal gland, accounting for 66% (244/372) of benign tumors and 38% (125/325) of malignant tumors. There were variations in the proportions of benign and malignant tumors at different minor salivary gland sites. The locations of benign tumors included 15% (55/372) on the upper lip, 10% (39/372) on the bucca, 3% (11/372) on the lower lip, 2% (8/372) on the floor of the mouth, and 2% (7/372) on the lingual and 1% (4/372) on the retro-molar areas, whereas those of malignant tumors accounted for 18% (57/325) on the floor of the mouth, 14% (44/325) on the bucca, 11% (35/325) on the tongue, 7% (22/325) retro-molar, 5% (17/325) on the upper lip and 4% (12/325) on the lower lip. Contrary to the major salivary gland tumors, most of the minor salivary gland tumors showed a female predominance (Table 2). In addition, the relative proportion of malignant tumors was inversely correlated to glandular size: 14% (489/3505) in the parotid gland, 23% (171/747) in the submandibular gland, 75% (24/32) in the sublingual gland, 34% (125/ 369) on the palate, 24% (17/ 72) on the upper lip, 52% (12/23) on the lower lip, 53% (44/83) on the bucca, 83% (35/42) on the tongue, 88% (57/65) on the floor of the mouth and 85% (22/26) in the retro-molar area. The malignant tumors from the sublingual gland and all minor salivary glands but the palate and upper lip outnumbered the benign ones (Table 2; Figure 2). Only three cases (0.9%, 3/325) of polymorphous adenocarcinoma were completely derived from minor salivary glands.

Over the 39-year period, the annually registered number of benign tumors showed an increase of about 10 times in males and about 3 times in females, whereas those of malignant tumors had increased slightly but remained nearly unchanged (Figure 5). The age-adjusted registration rates per year per 100,000 population for benign tumors had also increased from 1.7 (1973–1977) to 5.7 (2008–2011) in males and from 1.8 (1973–1977) to 4.0 (2008–2011) in females, whereas those for malignant tumors had remained almost unchanged with an upper limit of around 1.0 in both sexes. The average annual age-adjusted incidence rate per 100,000 population was 3.3 for benign tumors and 0.8 for malignant tumors (Appendix A).

### 3.2. Characteristics of Representative Histological Types

#### 3.2.1. PA

This was the most common benign tumor, accounting for 54% (2712/5015) of all salivary gland tumors, 68% (2712/3998) of benign tumors, 55% (2366/4318) of major salivary gland tumors and 50% (346/ 697) of minor salivary gland tumors. There was a female predominance for both major salivary glands (M/F: 879/1487) and minor salivary glands (M/F: 139/207). The mean and median ages from 7–93 years old were 48.3 and 48 years old, respectively. The highest incidences of major salivary gland tumors in males and females and in minor salivary gland tumors in males and females were in the fourth decade, the sixth decade, the fifth decade and the sixth decade of life, respectively. Unlike most other salivary gland tumors, PA showed a relatively wide range of distribution between children and elderly people. The most common site of PA was the parotid gland (66%, 1785/2712), followed by the submandibular gland (20%, 552/2712), palate (8%, 230/2712), upper lip (2%, 53/2712), and the buccal area (1%, 36/2712). The age-adjusted registration rates per year per 100,000 population for PA of major and minor salivary glands in males showed an approximately two-fold increase during the 39-year period of 1973–2011, whereas that in minor salivary glands in females had been gradually decreasing.

#### 3.2.2. Warthin Tumor

This second most common tumor accounted for 21% (1054/5015) of all salivary gland tumors and 26% (1054/3998) of benign tumors. This tumor, with its striking male predominance (M/F: 898/156), was exclusively derived from the parotid gland (99%, 1045/1054), and occurrence from other sites was extremely rare. The mean and median ages from 17 to 90 years old were 63.3 and 63 years old, respectively. The peak decade of incidence was the seventh decade for the males and the eighth decade for the females. Only 1% (15/1054) of Warthin tumors occurred before the age of 40 years, but three cases were derived from teenagers aged 13–19 years. The registration rates in males showed an approximately five-fold increase during the 39-year period, whereas that in the females had risen only slightly.

#### 3.2.3. ACC

ACC comprised 5% (271/5015) of all salivary gland tumors, 27% (271/1017) of malignant tumors, 3% (150/4318) of major salivary gland tumors and 17% (121/697) of minor salivary gland tumors. More female than male had ACC in both the major salivary glands (M/F: 55/95) and minor salivary glands (M/F: 50/71). The mean and median ages of the ACC from 14–96 years old were 59.4 and 61 years old, respectively, and the highest incidences of ACC were all in the seventh decade of life regardless of sex and location. The breakdown of locations was the parotid gland (28%, 76/271), submandibular gland (21%, 58/271), palate (18%, 48/271) and floor of mouth (10%, 27/271). ACC was the most predominantly malignant tumor among rare sublingual gland tumors (63%, 15/24). The age-adjusted registration rates per year per 100,000 population for ACC of major salivary glands had decreased slightly, whereas those of minor salivary glands showed a mild increase.

#### 3.2.4. MEC

MEC represented 5% (266/5015) of all salivary gland tumors, 26% (266/1017) of malignant tumors, 4% (157/4318) of major salivary gland tumors and 16% (109/697) of minor salivary gland tumors. Male outnumbered female (M/F: 88/69) for MEC from major salivary glands, but the opposite propensity was revealed in minor salivary glands (M/F: 47/62). The mean and median ages of the MEC from 12–97 years old were 57.4 and 60 years old. The highest incidences of both major and minor salivary gland tumors in the males and females were all in the seventh decade of life. MEC (47%, 8/17) was the most common malignant tumor in children and teenagers under 20 years of age. The most common site of MEC was the parotid gland (47%, 125/266), followed by the palate (13%, 34/ 266), submandibular gland (11%, 29/266), floor of the mouth (8%, 20/266), and buccal (5%, 14/266), lingual (5%, 14/266) and retro-molar regions (4%, 11/266). The number of MEC in the lower lip (*n* = 5) was the same as that in the upper lip (*n* = 5). The age-adjusted registration rates per year per 100,000 population for MEC of major salivary glands had not changed significantly, whereas those of minor salivary glands showed a tendency to increase.

#### 3.2.5. Carcinoma ex PA

This histology comprised 2% (111/5015) of all salivary gland tumors, 11% (111/1017) of malignant tumors, 2% (100/4318) of major salivary gland tumors and 2% (11/697) of minor salivary gland tumors. Male with these major salivary gland tumors outnumbered female (M/F: 65/35), although the opposite propensity was revealed for tumors in the minor salivary glands (M/F: 3/8). The mean and median ages from 17–89 years old were 61.4 and 63.0 years old, and the difference in each age between the carcinoma ex PA and PA was 13.1 years and 15 years, respectively. The highest incidences of this histology in major salivary gland were in the seventh decade of life. The most frequent site of occurrence was the parotid gland (63%, 70/111), followed by submandibular gland (23%, 26/111), minor salivary glands (10%; 11/111), and sublingual gland (4%; 4/111). The localization of involved minor salivary glands was predominated by the palate (73%, 8/11), followed by buccal (18%, 2/11) and floor of the mouth (9%, 1/11) locations. The age-adjusted registration rates per year per 100,000 population for males had increased, whereas those for females showed a decreasing tendency.

## 4. Discussion

The general features of salivary gland neoplasms from the Japanese population in the present study revealed some similarities and some differences compared with the previously reported literature from various parts of the world. The previously reported proportions of malignant salivary gland tumors ranged from 10–46% [9], and the present data showed a malignancy rate of 20% in Japan. That rate seemed to be slightly lower, but the actual percentage might be more or less, considering the transition of registered benign tumors. The proportions of benign and malignant tumors by percentage in the major and minor salivary glands were 72% for major benign, 14% for major malignant, 7% for minor benign, and 6% for minor malignant tumors in this study. Although the results were different from those of the AFIP Salivary Gland Registry [2], they appeared to be related to the types of cases referred to the reporting treatment centers. There were substantial fluctuations in age-standardized registration rates by the five-year calendar period and by histological type (Figure 6) due to the small number of cases.

The palate was the most common site for minor salivary gland tumors followed by the lip and buccal mucosa, as with the previous report from Japan [13]. Of the 95 labial salivary gland tumors, 72 (76%) were from the upper lip and 23 (24%) were from the lower lip, with 55 (76%) of the 72 upper lip tumors benign and 12 (52%) of the lower lip tumors malignant. These results showed the same tendency as the previous investigations [14], indicating that minor salivary gland tumors are more common in the upper lip than the lower lip, and the lower lip tumors are more likely to be malignant. As shown in the present study, the incidences of the representative histological types in Japan were almost similar to the previous reports from outside of Japan. However, the extremely rare occurrence (*n* = 3) of polymorphous adenocarcinoma was worthy of special mention, as with several reports from Japan [13,15], although all cases that had possibly been misdiagnosed as ACC or carcinoma ex PA were reviewed. Considering the different frequencies of polymorphous adenocarcinoma between Japanese and Chinese populations [9,16], the causative factors other than geographical and ethnic differences might be related. Further accumulation of carcinogenesis data will be needed.

Several studies have shown that there are considerable inter-observer variations in the assessment of histology between pathologists. In the present study, we re-evaluated the pathological diagnosis of almost all tumors, according to the latest WHO classification. SC is a new entity accepted in the latest WHO classification. Although SC was reported to present in the parotid gland of adults with equal sex distribution [1], our data showed that SC represented 2% of the malignant tumors with male predominance. Interestingly, four (17%) of the 23 cases arose in teenagers. Prior to the establishment of this entity, most SC were classified as AcCC, the second most common salivary gland malignancies in children.

The present study analyzed a total of 5015 salivary gland tumors using large-scale multi-institutional data registered in the HTTR over a period of 39 years in Hiroshima Prefecture, Japan. At the time of registration, all cases reported by local pathologists to the HTTR are confirmed to not already be registered in the database with use of personal identifying information such as name, date of birth, address, etc. Therefore, there is no duplication of cases, even if the tumor information is reported by different hospitals [17]. This registry has covered 75% of the cases registered in the Hiroshima Prefecture Cancer Registry, which is a population-based regional cancer registry and covers all institutions and hospitals in Hiroshima Prefecture [17]. However, the present study had several limitations. First, our data were based on tumors resected by surgery or biopsy and registered in the HTTR. Cases who did not undergo therapeutic or diagnostic resection were not registered in the database. Second, we observed 3998 benign salivary gland tumors over the 39 years; however, this is most likely an underestimation because the majority of cases with benign salivary gland tumors are not always transferred to cancer institutions or referral centers for problematic cases, and registration of those cases was deemed omitted. Indeed, the number of registered malignant tumors displayed increased slightly but remained nearly unchanged, whereas that of benign tumors increased by about three times. These increases were presumably influenced by the increment of small and medium sized hospitals or clinics. The number of participating institutions was 29 in 1973 at the start of registration, and it gradually increased over the 39 years to 85. In addition, the improvement of diagnostic accuracy must also have contributed to the increase of benign tumors that were asymptomatic and would not have been detected or resected for diagnosis without conventional radiological imaging methods, such as X-ray, computed tomography, and magnetic resonance imaging. Third, these data were not accompanied by other variables including tumor staging and comorbidity factors such as immunosuppression status, and any information on atomic bomb exposure or therapeutic radiation, although previous reports showed an increase in the annual incidence by these factors [18,19]. The absolute risk estimates of excess cases per 100,000 per year at 1 Sv were 3.7 for malignant tumors and 1.9 for benign tumors [20]. The population of interest in HTTR is different from the Life Span Study cohort of atomic bomb survivors followed by the RERF in Hiroshima, Japan [20,21]. Therefore, cases in the present study were less likely to be attributable to atomic bomb radiation. Since a national, population-based cancer registry or information from other prefectures with detailed and valid histological findings were unavailable, we believe that this study covering the whole areas of Hiroshima Prefecture is valuable and, in particular, the characteristics of malignant salivary gland tumors is close to the true perspective of Japanese population.

## 5. Conclusions

This investigation is the first, to our knowledge, to describe the relative frequency, characteristics and time trends of various salivary gland tumors in a Japanese population over a period of 39 years. We hope that the present data can contribute to the early diagnosis and effective treatment of salivary gland tumors and to cancer prevention.

## Figures and Tables

**Figure 1 jcm-08-00566-f001:**
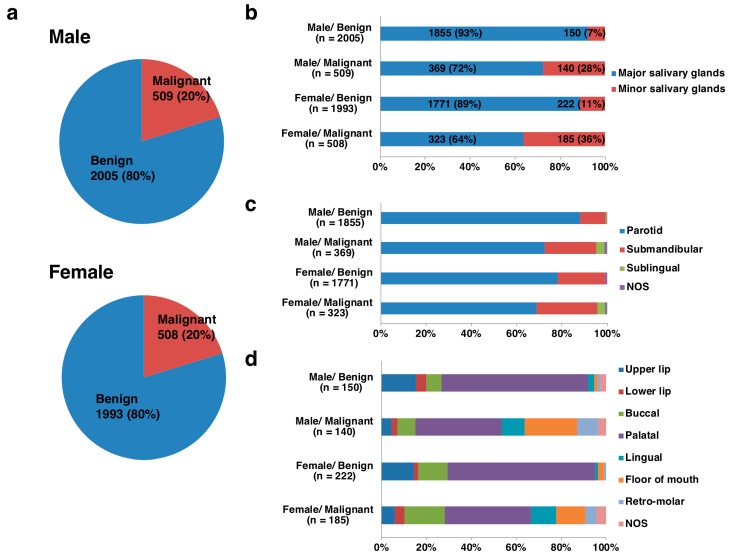
Breakdowns of benign and malignant tumors by sex and location in 5015 salivary gland tumors (**a**,**b**); 4318 major salivary gland tumors (**c**); and 697 minor salivary gland tumors (**d**). NOS: not otherwise specified.

**Figure 2 jcm-08-00566-f002:**
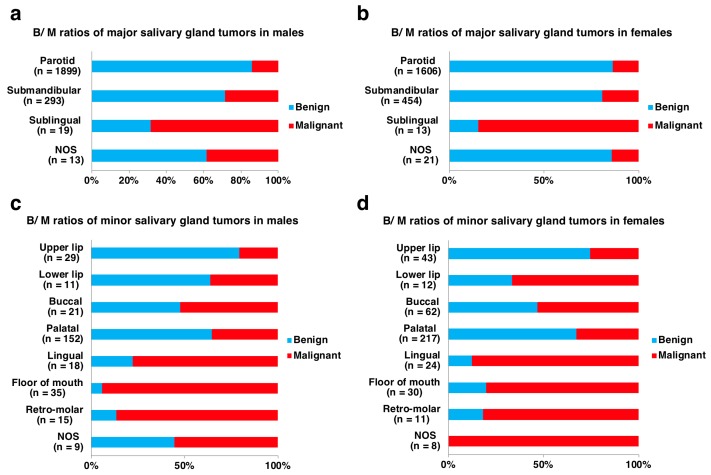
Breakdowns of benign and malignant tumors by sex and detailed location in 4318 major salivary gland tumors (**a**,**b**) and 697 minor salivary gland tumors (**c**,**d**). NOS: not otherwise specified.

**Figure 3 jcm-08-00566-f003:**
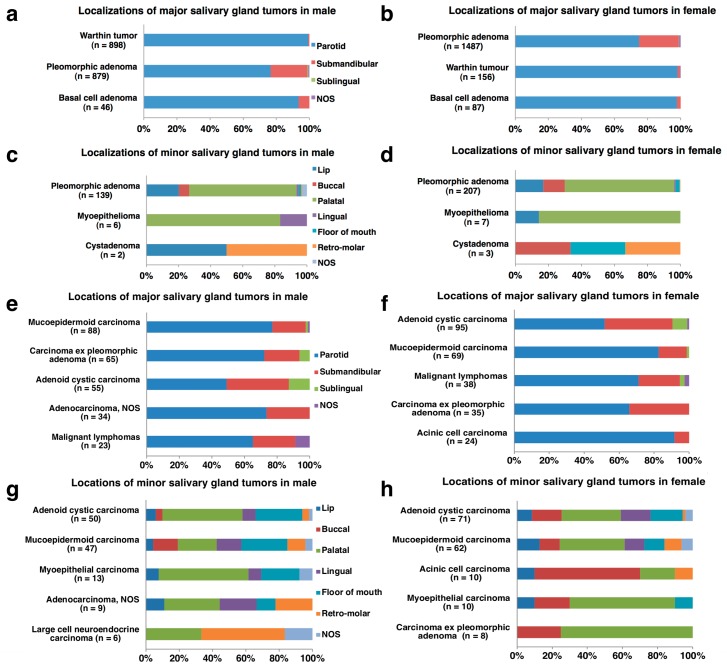
Representative histological frequencies and locations of major and minor salivary gland tumors (**a**–**h**). (**a**–**d**) benign salivary gland tumors, (**e**–**h**) malignant salivary gland; NOS: not otherwise specified.

**Figure 4 jcm-08-00566-f004:**
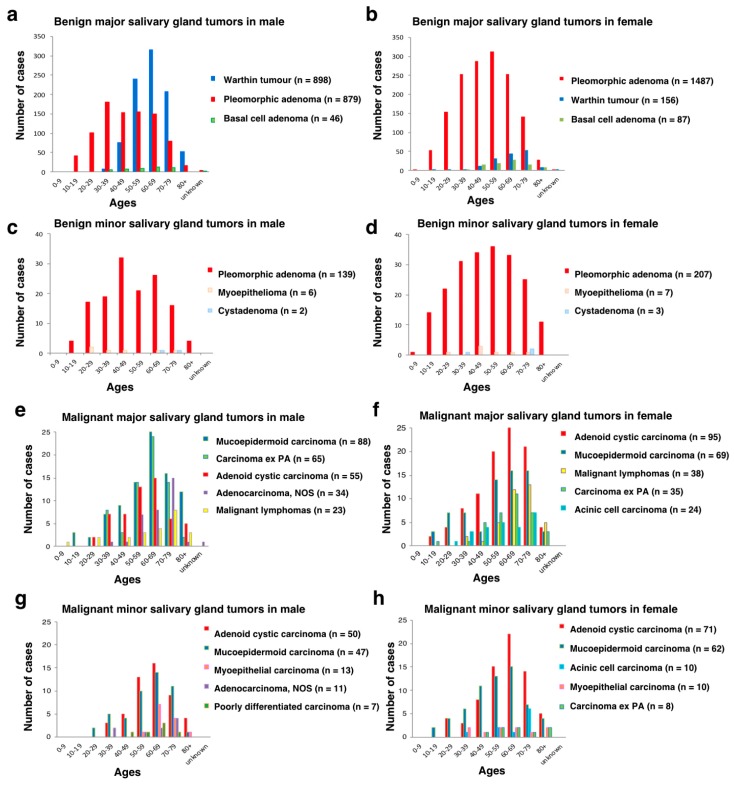
Age distributions of representative histologies in major and minor salivary gland tumors (**a**–**h**). (**a**–**d**) benign salivary gland tumors, (**e**–**h**) malignant salivary gland tumors; NOS: not otherwise specified; PA: pleomorphic adenoma.

**Figure 5 jcm-08-00566-f005:**
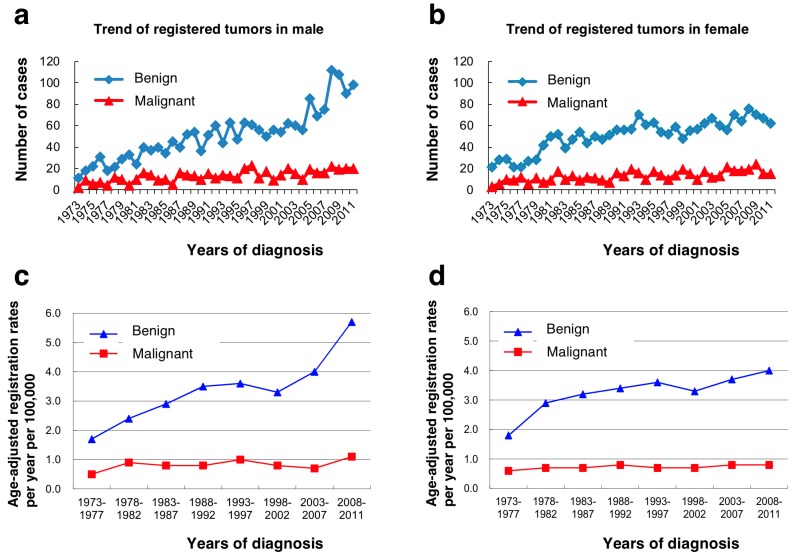
Annual registered number of cases (**a**,**b**) and age-adjusted registration rates per year per 100,000 population (**c**,**d**).

**Figure 6 jcm-08-00566-f006:**
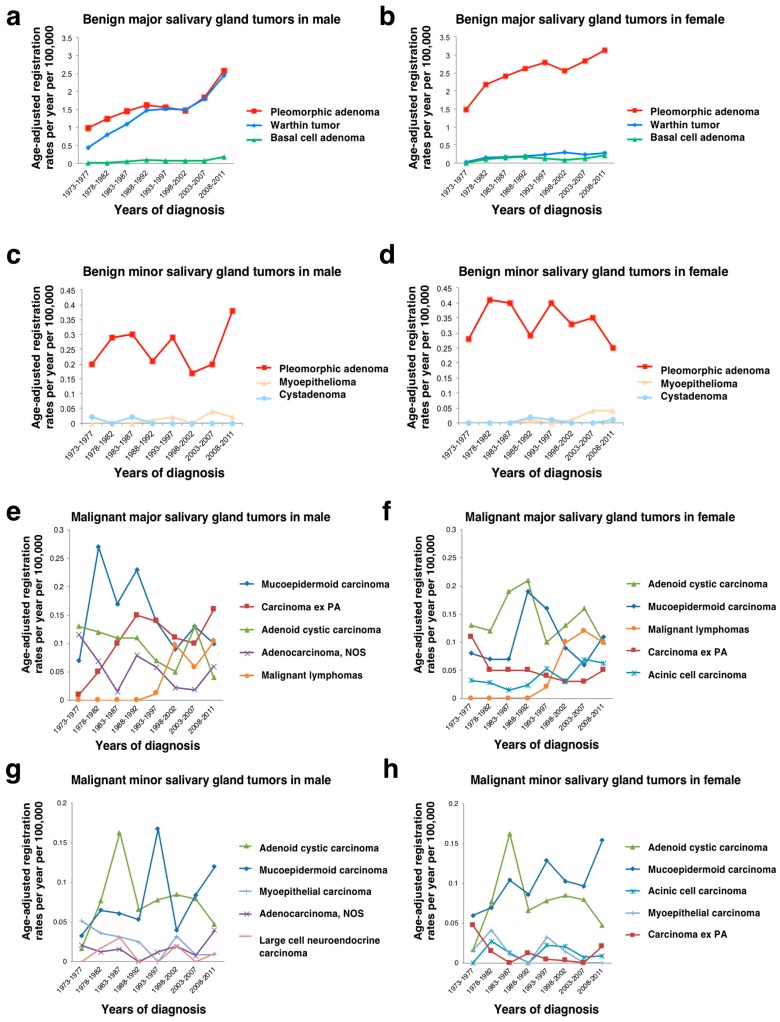
Age-adjusted registration rates per year per 100,000 population in major and minor salivary gland tumors. (**a**–**d**) benign salivary gland tumors, (**e**–**h**) malignant salivary gland tumors; NOS: not otherwise specified; PA: pleomorphic adenoma.

**Table 1 jcm-08-00566-t001:** Age and sex distribution in 5015 salivary gland tumors.

Age (years)	Benign		(M/F)	Malignant		(M/F)	Total	(B/M)
	M	F		M	F			
Major salivary glands (*n* = 4318)								
0–9	2	1	2	1	1	1	5	1.5
10–19	42	56	0.8	8	8	1	114	6.1
20–29	103	158	0.7	9	13	0.7	283	12
30–39	195	258	0.8	31	26	1.2	510	7.9
40–49	238	319	0.7	36	30	1.2	623	8.4
50–59	411	367	1.1	69	63	1.1	910	5.9
60–69	481	336	1.4	102	83	1.2	1002	4.4
70–79	307	221	1.4	77	78	1	683	3.4
80+	72	50	1.4	35	21	1.7	178	2.2
Unknown	4	5	0.8	1	0	-	10	-
Minor salivary glands (*n* = 697)								
0–9	0	1	0	0	0	-	1	-
10–19	4	14	0.3	0	2	0	20	9
20–29	19	23	0.8	2	8	0.3	52	4.2
30–39	20	33	0.6	11	12	0.9	76	2.3
40–49	34	37	0.9	11	25	0.4	107	2
50–59	21	38	0.6	30	36	0.8	125	0.9
60–69	29	35	0.8	48	49	1	161	0.7
70–79	18	30	0.6	30	37	0.8	115	0.7
80+	5	11	0.5	8	16	0.5	40	0.7
Total	2005	1993		509	508		5015	

Abbreviations: M: male; F: female; M/F: ratios of male to female; B/M: ratios of benign to malignant; NOS: not otherwise specified.

**Table 2 jcm-08-00566-t002:** Locations and sex distribution in 5015 salivary gland tumors.

Locations	Benign		(M/F)	Malignant		(M/F)	Total	(B/M)
	M	F		M	F			
Major salivary glands (*n* = 4318)								
Parotid	1632	1384	1.2	267	222	1.2	3505	6.2
Submandibular	209	367	0.6	84	87	0.9	747	3.4
Sublingual	6	2	3	13	11	1.2	32	0.3
NOS	8	18	0.4	5	3	1.7	34	3.3
Minor salivary glands (*n* = 697)								
Palatal	98	146	0.7	54	71	0.8	369	2.0
Upper lip	23	32	0.7	6	11	0.5	72	3.2
Lower lip	7	4	1.8	4	8	0.5	23	0.9
Buccal	10	29	0.3	11	33	0.3	83	0.9
Lingual	4	3	1.3	14	21	0.7	42	0.2
Floor of mouth	2	6	0.3	33	24	1.4	65	0.1
Retro-molar	2	2	1	13	9	1.4	26	0.2
NOS	4	0	-	5	8	0.6	17	0.3
Total	2005	1993		509	508		5015	4.0

Abbreviations: M: male; F: female; M/F: ratios of male to female; B/M: ratios of benign to malignant; NOS: not otherwise specified.

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
