# Peer review of "Characteristics of 5015 Salivary Gland Neoplasms Registered in the Hiroshima Tumor Tissue Registry over a Period of 39 Years"

_jcm, 2019, doi:10.3390/jcm8050566_

Reviewer 1 Report

This is a study that examines the incidence of salivary glands tumor in Hiroshima using registry data. The study is well written. There are multiple outcomes described in the study that could of interest for readers and future research. Comments:

 This study is not retrospective as there is no retrospective “follow-up” of cases. The entries of the study are cases. Each case was retrieved “cross-sectionally” at a given time point without follow-up data. it might be described as a retrospective review of the whole registry.

Please provide the ICD-O codes that were included. In Lines 77-78, the authors mention the grouping was based on ICD-O-3, however in line (64) the authors mention that records were coded using ICD-O-1, ICD-O-2, ICD-O-3. How did the authors classify entries using ICD-O-1 and ICD-O-2?.

Please clarify, was each specimen reviewed by both pathologists, or the specimens were divided between the pathologists? How many specimens were re-classified? In case the pathologists reviewed all the specimens, what was their agreement rate?

In the “Materials and methods”, please describe all the variables used in the study, statistical methods, and the statistical software used. I would suggest following the STROBE statement in organizing the “Materials and methods” section.

What other variables that database has that would let us understand more the risk of salivary tumor? Is there any information regarding the stage of malignancies? Comorbidity factors such as immunosuppression status? If no more information is available, please state that in the “limitations” paragraph.

I would suggest changing figure 4 from a line graph to a bar graph, the line graph gives the impression that the patients were followed until the diagnosis is made, a bar graph would be a better representation of the age categories created by the authors.

The discussion section is mostly addressing the incidence data, there is no or limited discussion of the trends line demonstrated in figures 4,5,6. In figure 6, it seems for the majority of malignancy types that the graph demonstrates significant fluctuations from year to year, this is unusual for a cancer incidence rate and unlikely to be explained by the improvement in diagnostic technologies, do the authors have an explanation for this, is it a limitation of the data or an actual fluctuation of cancer incidence?

While it is good to discuss the limitations of the study, it is advisable not to include it in the opening paragraph of the discussion, a good position is a paragraph before the conclusion.

Author Response

Reviewer 1: We thank the Reviewer 1 for general and specific comments. We have addressed all criticisms as follows.

 1.          The Reviewer 1 said that this study is not retrospective as there is no retrospective “follow-up” of cases, so we should describe it as a retrospective review of the whole registry.

 We agree fully with the opinion of the Reviewer 1. Thus, we have changed “a retrospective analysis” into “ a retrospective review” in Abstract (Page 2, line 3) of the revised manuscript.

 2.           The Reviewer 1 said that we should describe how we classified entries using ICD-O codes.

 We apologize for the insufficient descriptions. Thus, we have added the expression “Though tumors were originally coded according to the International Classification of Disease for Oncology (ICD-O), first, second, or third revision, all of the major and minor salivary gland tumors were re-grouped according to ICD-O-3 topography and morphology codes in the present study.” In Materials and Methods (Page 4, line 15-18).

 3.           The Reviewer 1 said that we should clarify the process of reviewing by both pathologists.

 We agree fully with the opinion of the Reviewer 1. Thus, we have added the expressions “All of 1017 malignant tumors were histologically reviewed and reclassified independently, by two pathologists (K. S. and I. O.) based on the latest (4th edition) of the World Health Organization (WHO) classification [1]. The concordance rate for diagnosis between two pathologists showed more than 80%, and when the evaluations differed, a final diagnosis was made by consensus while the investigators reviewed the specimen with a multi-head microscope.” in Materials and Methods (Page 4, line 18-22) of the revised manuscript.

 4.           The Reviewer 1 said that we should describe all the variables used in the study, statistical methods, and the statistical software used.

 We agree fully with the opinion of the Reviewer 1. Thus, we have added the expressions “We described the number and the proportion of cases by behavior (benign or malignant), sex, site of tumor occurrence, histological type, period at diagnosis (in 5-year groups), and age of diagnosis (in 10-year groups). We presented temporal trends (5-year calendar period) in sex-specific age-standardized registration rate per 100,000 population (standardized by Segi’s World Standard Population) by behavior (benign or malignant) and histological type. No formal statistical tests were performed and we used Microsoft Excel 2010.” in Materials and Methods (Page 5, line 4-9) of the revised manuscript.

 5.           The Reviewer 1 said that we should describe about other variables or any information about tumor staging and comorbidity factors.

 We agree fully with the opinion of the Reviewer 1, but no more information is available. Thus, we have added the expression “these data were not accompanied by other variables including tumor staging and comorbidity factors such as immunosuppression status” in the limitations paragraph (Page 13, line 9-10).

 6.           The Reviewer 1 recommended that we should change Fig. 4 from a line graph to a bar graph.

 We agree fully with the opinion of the Reviewer 1, and we changed a line graph to a bar graph in Fig. 4.

 7.           The Reviewer 1 said that we should explain about the fluctuations of tumors’ registration rates from year to year in Fig. 6.

 We agree fully with the opinion of the Reviewer 1. Thus, we have added the expression “There were substantial fluctuations in age-standardized registration rates by 5-year calendar period and by histological type (Fig. 6) due to small number of cases.” In Discussion (Page 11, line 14-15) of the revised manuscript.

 8.           The Reviewer 1 recommended that we should move the limitations paragraphs before the conclusion.

 We agree fully with the opinion of the Reviewer 1. Thus, we transferred the limitations paragraphs before the conclusion in the revised manuscript.

Reviewer 2 Report

The study entitled “Characteristics of 5015 Salivary Gland Neoplasms registered in the Hiroshima Tumor Tissue Registry over a period of 39 years “, is interesting, well-presented, and with implications for clinicians. Besides, I have highlighted several points in which this manuscript needs to be improved.

1.      In the methods section, line 73, the word “subjects” is considered discriminatory. Please replace with the word “patients” or “participants”.

2.      Table 2, the words in the column of location should be aligned to the left for better readability.

3.      In the discussion section, line 258, this registry has covered “almost all” institutions and hospitals in Hiroshima Prefecture. Please provide the definite number or ratio about how many institutions were covered.

Author Response

Reviewer 2: We thank the Reviewer 2 for the critical comments that have helped us to improve the manuscript. We have addressed all criticisms as follows.

 1.           The Reviewer 2 suggested that we should replace the word “subjects” with the word “patients” or “participants”.

 We agree fully with the opinion of the Reviewer 2. Thus, we replaced “subjects” with “patients” in Materials and Methods (Page4, line 11) of the revised manuscript.

 2.           The Reviewer 2 suggested that we should align the words in the column of location to the left in Table 2.

 We agree fully with the opinion of the Reviewer 2. Thus, we changed that way in Table 2.

 3.           The Reviewer 2 said that we should describe the definite number or ratio about how many institutions were covered in this registry.

 We agree fully with the opinion of the Reviewer 2. Thus, we have added the expression “This registry has covered 75% of the cases registered in the Hiroshima Prefecture Cancer Registry, which is a population-based regional cancer registry and covers all institutions and hospitals in Hiroshima Prefecture [17].” In Discussion (Page 12, line 18-20) of the revised manuscript.

Round  2

Reviewer 1 Report

The authors have addressed the comments sufficiently